Accepted at the ICLR 2024 Workshop on AI4Differential Equations In Science

# LEARNING ITERATIVE ALGORITHMS TO SOLVE PDES

**Lise Le Boudec** [1] [*]    **Emmanuel De Bézenac** [2] [*]    **Louis Serrano** [1]

**Yuan Yin** [1]    **Patrick Gallinari** [1,3]

[1] Sorbonne Université, CNRS, ISIR, F-75005 Paris, France
[2] Seminar for Applied Mathematics, ETH, Zurich, Switzerland
[3] Criteo AI Lab, Paris, France

## ABSTRACT

In this work, we propose a new method to solve partial differential equations (PDEs). Taking inspiration from traditional numerical methods, we view approximating solutions to PDEs as an iterative algorithm, and propose to learn the iterations from data. With respect to directly predicting the solution with a neural network, our approach has access to the PDE, having the potential to enhance the model's ability to generalize across a variety of scenarios, such as differing PDE parameters, initial or boundary conditions. We instantiate this framework and empirically validate its effectiveness across several PDE-solving benchmarks, evaluating efficiency and generalization capabilities, and demonstrating its potential for broader applicability.

## 1 INTRODUCTION

Partial Differential Equations (PDEs) are ubiquitous as mathematical models of interesting phenomena in science and engineering (Evans, 2010). Traditional approaches to solving PDEs involve numerical methods such as finite difference and finite element analysis (Quarteroni & Valli, 1994). These methods iteratively refine an initial solution towards greater accuracy, employing iterative solvers like Jacobi, Gauss-Seidel, and Krylov subspace methods. Given the complexity and ill-conditioned nature of many PDEs, these iterative processes can demand extensive computational resources. Preconditioning techniques are often essential to mitigate this, though they require precise customization to the specific PDE problem, making the development of effective solvers a significant research endeavor in itself. Yet, the computational demands, time and expertise required to develop these algorithms sometimes make them impractical, leading to an interest in machine learning (ML) based alternatives (Karniadakis et al., 2021).

To date, ML-based approaches to solving PDEs have predominantly fallen into two categories: *supervised* and *unsupervised*. The *supervised* methodology consists in first solving the PDE using numerical methods to generate input and target data, and then regressing to the solution using neural networks. Many models, such as Neural Operators, lie within this class (Li et al., 2020; Raonić et al., 2023; Bartolucci et al., 2023) and focus on learning the solution operator directly through a single neural network pass. This method is very efficient, at the downside of relying on quite large quantities of data for training in order to ensure adequate generalization. Additionally, the neural network does not have access to the PDE is in itself never used, only indirectly through the data.

The *unsupervised* approach, involves considering a neural network as solution of the PDE, whose parameters are found by minimizing the PDE residual with some form of gradient descent. Methods such as Physics-Informed Neural Networks (PINNs) (Raissi et al., 2019), or DeepRitz (E & Yu, 2018) fall under this category. This method is attractive as it does not rely on any form of data, relying on information from the PDE residual instead. This means that is holds the promise to generalize to any PDE, initial, and boundary conditions. However, this model exhibits difficulties during training (Krishnapriyan et al., 2021; Ryck et al., 2023), often requiring many optimization steps and sophisticated training schemes (Krishnapriyan et al., 2021).

In this work, we introduce a novel class of neural network-based PDE solvers, designed to synergize the strengths of both supervised and unsupervised methodologies. Our approach consists in *learning an iterative algorithm that solves the PDE*. The hope is that by learning the algorithm rather that directly predicting the solution as in the supervised methods, we obtain improved generalization, and a smaller number of iterations w.r.t. unsupervised approaches.

## 2 ITERATIVE NEURAL PDE SOLVERS

### 2.1 PROBLEM STATEMENT

Let us consider the following family of boundary value problems parametrized by $\gamma$ with domain $\Omega$, potentially representing both space and time, with $\mathcal{N}$ a potentially nonlinear differential operator, $\mathcal{B}$ the boundary operator, $g$ the initial/boundary conditions and source term $f$:

$$\mathcal{N}(u; \gamma) = f \quad \text{in } \Omega, \tag{1}$$
$$\mathcal{B}(u) = g \quad \text{on } \partial\Omega. \tag{2}$$

Note that different PDEs can be represented in this form, amounting to changing the parameters $\gamma$. The goal here is to develop a generic algorithm that is able to solve the above problem, yielding an approximate solution $u$ given the PDE and different sets of inputs $(\gamma, f, g)$.

We assume access to a dataset of $M$ samples: for each set of inputs $(\gamma_i, f_i, g_i)_{i=1}^M$ the associated target solution $(u_i)_{i=1}^M$ given on a $m$ point grid $(x_j)_{j=1}^m$. These observations will be used during training, in order to learn the solver. During inference, only the inputs related to a new PDE instance are given.

### 2.2 GENERAL METHODOLOGY

Traditionally, numerical solvers consider an ansatz $u_\theta$ parametrized by some finite dimensional $\theta$, found by iteratively [1] updating parameters $\theta$ based on the minimization of some criterion $\mathcal{L}_{\text{PDE}}$ (e.g. the PDE residual) , which assesses how well $u_\theta$ meets the conditions specified in equations 1 and 2.

We take inspiration from traditional methods and propose a data-driven, iteration based algorithm that leverages $\mathcal{L}_{\text{PDE}}$ in order to make informed updates. As opposed to traditional methods, the iterative solver $\mathcal{A}$ is not specifically tailored or handcrafted to the given problem, but *learned* from the data. The updates take the following form, starting from an initial $\theta_0$:

$$\theta^{k+1} = \mathcal{A}(\theta^k; \mathcal{L}_{\text{PDE}}, \gamma, f, g) \tag{3}$$

The primary goal is to iteratively refine the ansatz, $u_K := u_{\theta_K}$, to closely approximate the true solution after a series of $K$ iterations, ideally small for efficiency. The algorithm is trained from input target data from different sets of PDE parameters $\gamma$, sources $f$, initial and boundary conditions $g$ as outlined in section 2.1. The underlying hypothesis is that even though the solutions may be different for different inputs and parameters, the underlying solution methodology remains relatively consistent. This consistency is expected to enhance the algorithm's ability to generalize across novel scenarios effectively.

**Comparison w.r.t. to supervised and unsupervised approaches.** This framework is somewhat different from both classes of methods discussed in the introduction: With respect to supervised approaches, instead of directly learning to predict the solution, we learn the iterative algorithm that solves the PDE. As it has access to the PDE, the hope is that it is able to assess whether the ansatz is close to the solution and refine its predictions in the following steps, as opposed to one step. Instead of learning to predict the solution, it learns to solve the PDE. Our intuition is that the proposed algorithm may require less data w.r.t. these methods as it has access to the PDE. With respect to unsupervised approaches where $\mathcal{A}$ would correspond to the optimization algorithm e.g. SGD or Adam. Our updates are learned using a neural network, thus requiring much less iterations.

---

[1]Other standard methods include time marching schemes, such as Euler Forward, can be seen as iterative methods.

### 2.3 A Possible Instantiation

Below, we present an instantiation of the framework selected for the experiments described in section section 3.

**Choice of Ansatz** $u_\theta$. A very common choice (Shen et al., 2011) is to consider a family of basis functions $\Psi(x) = \{\psi_i(x)\}_{i=1}^N$ and consider the ansatz to be given by its linear span $u_\theta(x) = \sum_{i=0}^N \theta_i \psi_i(x)$. In the following we consider this linear reconstruction, although our formulation is generic in the sense that it can also accommodate nonlinear variants[2].

**Choice of criteria** $\mathcal{L}_{\text{PDE}}$. Similar to PINNs(Raissi et al., 2019), we consider $\mathcal{L}_{\text{PDE}}$ to be given by the strong formulation of the residual $\mathcal{L}_{\text{Res}}$, plus a boundary discrepancy term $\mathcal{L}_{\text{BC}}$[3].

$$\mathcal{L}_{\text{PDE}} = \mathcal{L}_{\text{Res}} + \lambda \mathcal{L}_{\text{BC}}, \quad \mathcal{L}_{\text{Res}} = \sum_{x_j \in \Omega} |\mathcal{N}(u_\theta; \gamma)(x_j) - f(x_j)|^2, \quad \mathcal{L}_{\text{BC}} = \sum_{x_j \in \partial\Omega} |\mathcal{B}(u_\theta)(x_j) - g(x_j)|^2$$

**Choice of update algorithm** $\mathcal{A}$. A simple instantiation of the iterative solver can be obtained, inspired by gradient based approaches. Given the parameters $\theta_k$ of the ansatz at iteration $k$, the steepest direction of loss $\mathcal{L}_{\text{PDE}}$ is computed, using autograd. The gradient is then transformed with a neural network $\mathcal{F}_\varrho$ with parameters $\varrho$, depending on the values of the PDE parameters as well as the other inputs.

$$\theta^{k+1} = \theta^k - \eta \mathcal{F}_\varrho(\nabla \mathcal{L}_{\text{PDE}}(\theta^k), \; \gamma, f, g) \tag{4}$$

This process bears some similarities with the concept of preconditioning in traditional numerical analysis (Benzi, 2002). If the neural network were to function as an identity, this method would simplify to a form of Stochastic Gradient Descent (SGD) akin to that used within Physics-Informed Neural Networks (PINNs). Considering the often ill-conditioned nature of PDE training processes, a direct application without adaptation would lead to prohibitive iteration counts for convergence (Ryck et al., 2023), as demonstrated in section 3. By modulating the gradient with $\mathcal{F}_\varrho$ trained on input-solution pairs, the aim is to reduce the number of required steps for a given accuracy. The network $\mathcal{F}_\varrho$ is trained to minimize the $l_1$ distance between the final ansatz $u_K$ of eq. (4) and the associated solution. Technical details are presented in appendix B.

## 3 Experimental Evaluation

To evaluate our approach, we perform a series of experiments on various PDEs including the Helmholtz and Poisson equations in 1D, along with advection equations, and extend our assessment to the 2D Darcy flow problem, as detailed in appendix C. Following the framework outlined in section 2.1, we train our model using pairs of inputs and corresponding solutions. The effectiveness of our method is then tested across diverse scenarios by altering PDE parameters, boundary conditions, initial states, and forcing terms.

For *Helmholtz*, our experiments involve varying wave numbers, chosen uniformly within a specified range, and altering boundary conditions. For *Poisson*, the inputs correspond to the forcing function and the values at the boundary. The *advection* problem is taken from the PDE benchmark (Takamoto et al., 2023), where different advection speeds and initial conditions are considered. For the *Darcy* problem from (Li et al., 2020), diffusion coefficients are varied. For comprehensive details on these benchmarks, please refer to section appendix C. We compare our approach in 2 settings. First, we assess its generalization capabilities by comparing it to classical DL methods with trainings and then, we show the convergence speed at test-time optimization w.r.t. solvers.

### 3.1 Evaluating Generalization Performance

We first compare our method against both supervised and unsupervised approaches. *NN*: a supervised neural network trained to regress directly to the solution, *PINNs*: a PINNs where the input

---

[2]Although in our preliminary results we have found this may further complicate training.

[3]Note that more generic formulations of the loss may also be considered in a straightforward manner.

data is fed to the network, by taking inspiration from (Zhang et al., 2023). Subsequently, we present two baselines with hybrid training approaches, using both the data and the PDE: Physics-Informed DeepONet (*PIDON*) (Goswami et al., 2022) and Physics-Informed Neural Operator (*PINO*) (Li et al., 2023). Hybrid training considers using both data through a supervised setting and PDE by adding a physical loss. *PINO* closely aligns with our approach as it incorporates a hybrid training phase and unsupervised test-time optimization based on $\mathcal{L}_{\text{PDE}}$. However, the second stage is simply optimization, not like ours which has to learn to optimize. See appendix A for a more detailed literature review and appendix B for training details. We provide a comparison of the benefits of the training methods for the aforementioned models in table 1 and qualitative comparison in appendix D.

| | Baseline | 1d | | 1d + time | 2d |
|---|---|---|---|---|---|
| | | Helmholtz | Poisson | Advection | Darcy-Flow |
| Unsupervised | *PINNs* | 4.26e-1 | 2.33 | 2.26e-1 | 2.31e-1 |
| Supervised | *NN* | 9.04e-2 | 1.09e-1 | 1.27e-1 | 7.54e-3 |
| | *PIDON* | 4.67e-1 | 8.84e-2 | 2.26e-1 | 4.46e-2 |
| Hybrid | *PINO* | 4.78e-1 | 5.83e-3 | **1.83e-4** | 2.80e-2 |
| | *Ours* | **1.10e-2** | **1.76e-3** | 8.40e-3 | **6.21e-3** |

Table 1: Results - metrics in MSE on the test set.

In Table 1, we can observe that all other methods using the PDE (*PINNs*, *PIDON*, *PINO*) seem to face challenges in learning effectively from the physical loss alone (apart for the advection problem, where Fourier layers seem to be particularly adapted (Takamoto et al., 2023)). This may be attributed to the inherent difficulties associated with the conditioning of the physical loss, denoted as $\mathcal{L}_{\text{PDE}}$, which poses significant challenges for models relying primarily on physical constraints. In contrast, our approach appears to adeptly navigate these challenges, effectively learning to bypass the limitations imposed by the physical loss. Moreover, when compared to the purely supervised baseline, our model exhibits superior generalization performance.

## 3.2 Evaluating Computational Efficiency

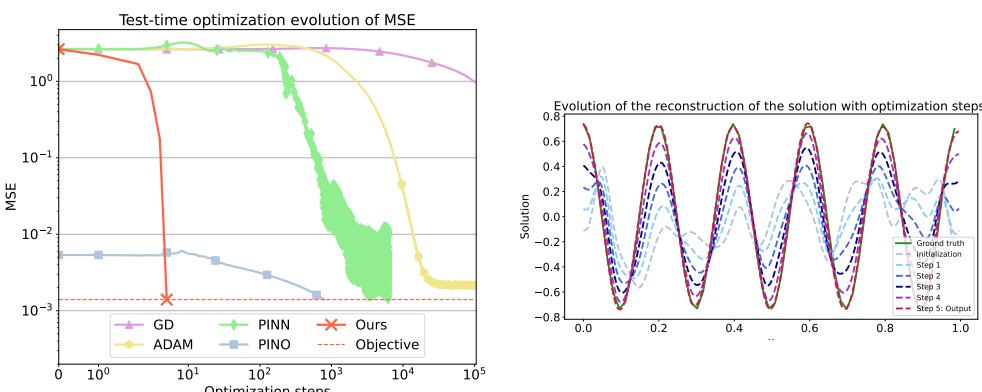

(a) Error vs number of iterations (log-scale). Rela-tive to other iteration-based approaches, our method demonstrates significantly faster performance (Poisson).

(b) Qualitative behavior of our approach at each itera-tion. We can observe the method gradually converges towards the target solution (Helmholtz).

We compare our learned optimizer against other unsupervised approaches for test-time optimization, considering access solely to the PDE and its parameters for solving. We evaluate various iterative algorithms, including gradient-based optimizers (SGD and Adam (Kingma & Ba, 2015)) and DL solvers: Physics-Informed Neural Network (PINN) and Physics-Informed Neural Operator (PINO). Specifically, we compare test-time optimization speed in a fully unsupervised setting, where only $\mathcal{L}_{\text{PDE}}$, $\gamma$, $f$, and $g$ are accessible. For the PINO baseline, we utilize the pre-trained model from section 3.1 and fine-tune it on $\mathcal{L}_{\text{PDE}}$ as specified in (Li et al., 2023).

Figure 1a and fig. 1b show qualitative results for the **test-time optimization** steps on the physical loss $\mathcal{L}_{\text{PDE}}$. While, baselines have difficulties in the optimization due to the ill-conditioning of the physical loss and show a slow convergence, NGD goes directly to a high quality solution in 5 steps (fig. 1a and appendix D). PINO (blue) has a better initialization than other models, thanks to its pre-training phase, but still has slow convergence speed because it performs an unsupervised fine-tuning. Indeed, on fig. 1a, PINO and PINN required 793 and 6307 fine-tuning steps respectively to reach our solver's accuracy, while ADAM and GD did not make it within the $100,000$ steps.

## 4  CONCLUSION

In this work, we proposed a PDE solver based on an iterative process, highlighting its potential for generalization performance and efficiency. While the initial findings are promising, they are also preliminary. Moving forward, we aim to explore more sophisticated architectures for the solver, such as convolutional neural networks or attention-based neural networks, and to extend our analysis to include datasets in two or three dimensions.

### ACKNOWLEDGMENTS

We acknowledge the financial support provided by DL4CLIM (ANR-19-CHIA-0018-01), DEEP-NUM (ANR-21-CE23-0017-02), PHLUSIM (ANR-23-CE23-0025-02), and PEPR Sharp (ANR-23-PEIA-0008", "ANR", "FRANCE 2030").

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

APPENDIX

# A  RELATED WORK

**Solvers and Deep Learning**: Many tools for solving numerically PDE have been developed for years. The standard methods for PDE include Finite Differences Method (FDM), Finite Volume Method (FVM), Finite Element Method (FEM), spectral methods, multigrid methods and many others (S. H, 2012; Liu, 2009). While these methods are widely used, they often suffer from a high computational cost for complex problems or high-precision simulations. To address these challenges, integrating deep learning (DL) into solvers has emerged as a promising approach. Current solutions include incorporating correction terms to reduce numerical errors (Um et al., 2021; Hsieh et al., 2019), or leveraging meta-learning to discover optimal initializations (Qin et al., 2022). Algortihm unrolling is also used to cascade iterations and improve performances of iterative algorithms (Monga et al., 2020). This approach is different from ours since we keep the iterative part of the solver and aim at improving each iteration. Lastly, learned optimizers are models used to accelerate optimization scheme. If this literature is close to our model, it has not yet been directly applied to PDE solving from physical data (parameters, boundary conditions...) (Chen et al., 2021).

**Unsupervised training**: Physics-Informed Neural Networks (PINNs) (Raissi et al., 2019) have been a pioneering work in the development of DL method for physics. In these models, the solution is a neural network that is optimized using the residual loss of the PDE being solved. However, this method suffers from several drawbacks. First, as formulated in (Raissi et al., 2019), a PINNs can solve one instance of an equation at a time. Any small change in the parameters of the PDE involves a full re training of the network. Efforts such as (Beltrán-Pulido et al., 2022; Zhang et al., 2023) have attempted to address this limitation by introducing parametric versions of PINNs capable of handling parametric equations, while (Huang et al., 2022) explores meta-learning approaches. Moreover, PINNs have shown convergence difficulties: (Krishnapriyan et al., 2021) show that PINNs' losses have complex loss-landscapes, complicating training despite adequate neural network expressiveness. Approaches like those detailed in (Wang et al., 2022) adopt a Neural Tangent Kernel (NTK) perspective to identify reasons for failure and suggest using adaptive weights during training to enhance performance. Additionally, studies such as (Ryck et al., 2023) demonstrate that PINNs suffer from ill-conditioned losses, resulting in slow convergence of gradient descent algorithms.

**Supervised trainings**: In contrast to the unsupervised training of Physics-informed Neural Networks, purely data-driven models have demonstrated remarkable capabilities for PDE simulation and forecasting. In most of the existing literature, the entire solver is replaced by a DL architecture, and focused on directly computing the solution from a given input data. A widely studied setting is operator learning which learns mappings between function spaces (Li et al., 2020; Kovachki et al., 2023; Lu et al., 2021).

**Hybrid models**: Hybrid models are models that use both the available physical knowledge and data. Some example include the Aphinity model (Yin et al., 2021) (where the author have partial knowledge on the physics and learned the remaining dynamics from data), Physics-informed Deep Operator Networks (PIDON) (Wang et al., 2021; Goswami et al., 2022), Physics-informed Neural Operator (PINO) (Li et al., 2023) (DeepONet architecture (Lu et al., 2021) or Neural Operator models (Kovachki et al., 2023; Li et al., 2020) respectively with a combination of data and physical losses).

# B  IMPLEMENTATION DETAILS

We add here more details about the implementation and experiments presented in section 3.

## B.1  B-SPLINE BASIS

We chose to use a B-Spline basis to construct the solution. We manually build the spline and compute its derivatives thanks to the formulation and algorithms proposed in (Piegl & Tiller, 1996). We used Splines of degree $d = 3$ and constructed the Splines with 2 different ways:

- Take $N + d + 1$ equispaced nodes of multiplicity $1$ from $\frac{d}{N}$ to $1 + \frac{d}{N}$. This gives a smooth local basis with no discontinuities (see Figure 2a) represented by a shifted spline along the x-axis.
- Use $N + 1 - d$ nodes of multiplicity $1$ and $2$ nodes of multiplicity $d$ (typically on the boundary nodes: $0$ and $1$). This means that nodes $0$ and $1$ are not differentiable (see Figure 2b).

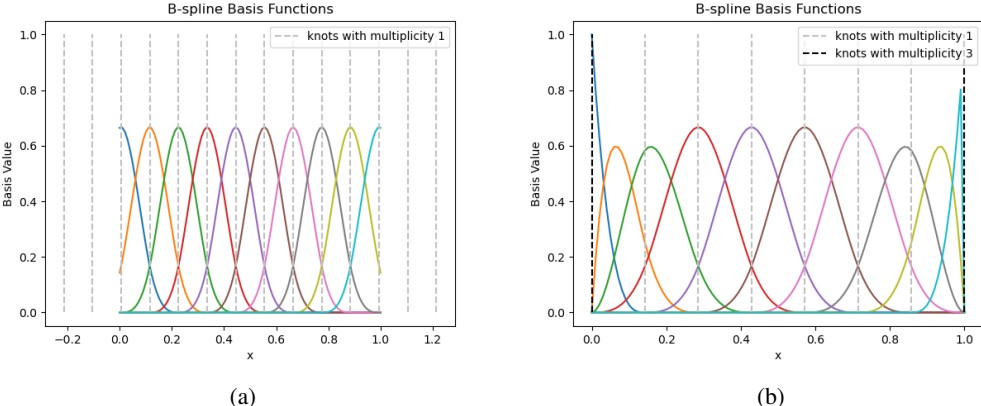

(a)                (b)

Figure 2: B-spline basis with $N = 10$ terms with shifted spline (Left) and higher multiplicity nodes (Right). Dashed lines represents nodes position with color the darker, the higher the multiplicity.

## B.2 TRAINING DETAILS

In our experiments, neural networks are trained using the Adam optimizer (Kingma & Ba, 2015). For network optimization, we employ a smooth $l1$-loss with a variable threshold for our solver while for other baselines, we use MSE loss and/or physical losses. All models are trained for at least $400$ epochs on datasets composed of some sampling of $\gamma$ and/or $f$ and/or $g$.

| | | Dataset | | | |
| | | 1d | | 2d | |
| Type | Model | Helmholtz | Poisson | Advection | Darcy-Flow |
|---|---|---|---|---|---|
| Model-driven | pPINNs | 0h10 | 0h05 | 0h20 | 1h45 |
| Data-driven | Dd | 0h40 | 0h05 | 0h20 | 0h15 |
| Hybrid Training | PIDON | 0h10 | 0h05 | 1h45 | 5h15 |
| | PINO | 0h05 | 0h15 | 1h30 | 1h45 |
| | N GD (us) | 0h40 | 0h20 | 19h20 | 21h15 |

Table 2: Training time for each baseline and dataset.

## B.3 BASELINES DETAILS

If not stated otherwise, all neural networks are 2 layers MLP with 256 neurons (ours included) and GeLU activation function (Hendrycks & Gimpel, 2016), but we had to make some hyper parameter research on some baselines.

For the parametric PINN model, data-driven training and Physics-informed DeepONet, we empirically searched hyper parameter to allow network to handle the high frequencies involved in the dataset. We used 5 layers and 200 neurons on the (Helmholtz) dataset for the parametric PINNs baselines, but kept the same network size as ours for other models, since bigger network sizes did not improves the results. On (Darcy) and (Advections), Physics-Informed DeepONet are trained

| Type | Model | Dataset | | | |
| | | 1d | | 2d | |
| | | Helmholtz | Poisson | Advection | Darcy-Flow |
|---|---|---|---|---|---|
| Optimizer | GD | 101 (0.36) | 67 (2.2) | 978 (1.11) | 2034 (0.33) |
| | ADAM | 92 (0.35) | 57 (0.04) | 562 (1.17) | 1572 (0.35) |
| Pre-trained | PINO | 270 (0.35) | 21 (1.4e-3) | 024 (4.36e-4) | 282 (1.23e-2) |
| Physics-Informed | PINNs | 149 (0.25) | 32 (1.3e-3) | 177 (0.68) | 2688 (0.27) |
| | N GD (us) | 0.04 (1.61e-3) | 0.03 (1.4e-3) | 0.27 (1.96e-2) | 0.25 (8.16e-3) |

Table 3: Test-time optimization duration (in seconds) for each baseline and dataset for solving a new equation instance (i.e. with new parameters). We let $10,000$ steps for baseline and convergence criteria is fixed to the error reached by our model within its $5$ steps. We add the final error after the $10,000$ steps (in brackets).

with 5 layers, 256 units in both branch net and trunk net because the dimension of these dataset are higher.

For the PINO baseline, we based the network architectures on those proposed by the author and kept a layer width of $64$. For $1d$-datasets, we used 2 FNO layers with 10 modes, except for (Helmholtz) for which we used 32 modes to let high frequencies being reconstructed. For the Darcy and Advection datasets, we used 4 FNO layers with 20 modes on each dimension (x, y) for (Darcy) and 20 and 12 modes for x and t respectively for (Advection). This leads to bigger networks than our ($\times 10$ more parameters for Advections and Darcy). See table 4, for a comparison of the number of parameters involved in each baseline.

| Type | Model | Dataset | | | |
| | | 1d | | 2d | |
| | | Helmholtz | Poisson | Advection | Darcy-Flow |
|---|---|---|---|---|---|
| Model-driven | pPINNs | $1,316,353$ | $151,050$ | $\underline{1,116,673}$ | $\underline{1,116,673}$ |
| Data-driven | Dd | $\mathbf{140,832}$ | $148,993$ | $1,547,664$ | $1,547664$ |
| Hybrid Training | PIDON | $664,597$ | $285,717$ | $1,648,149$ | $1,648,149$ |
| | PINO | $279,233$ | $\mathbf{99,009}$ | $13,140,737$ | $13,140,737$ |
| | N GD (us) | $\underline{149,024}$ | $\underline{137,482}$ | $\mathbf{993,680}$ | $\mathbf{993,680}$ |

Table 4: Number of parameters for each baselines and dataset. Bold shows the smallest model for a given dataset. We underline the second smallest model.

Table 4 shows that the baselines considered in section 3 involved network of similar sizes than ours or bigger. Since the architecture (number of layers and width) are comparable (except for PINO), the difference of weight's number is explained by the input and output sizes. As an example, our solver inputs $\nabla \mathcal{L}_{\text{PHY}}$ and some parameters $\gamma, f, g$, and outputs a descent direction for the parameters $\theta$, while PINNs also inputs the coordinate $x$ and outputs the solution on that query points $u(x)$ and NN predicts the coefficient $\theta$ in the basis.

## C    DATASET DETAILS

### C.1    HELMHOLTZ

We generate a dataset following the $1d$ static Helmholtz equation eq. (5). For $x \in [0, 1[$,

$$
\begin{cases}
\frac{\partial^2 u(x)}{\partial x^2} + \omega^2 u(x) & = 0 \\
u(0) & = u_0 \\
\frac{\partial u(0)}{\partial x} & = v_0
\end{cases}
\tag{5}
$$

The solution can be computed with : $u(x) = \alpha \cos(\omega x + \beta)$, with $\beta = \arctan(\frac{-v_0}{\omega u_0})$, $\alpha = \frac{u_0}{\cos(\beta)}$ are directly computed from the PDE data. We generate $1,000$ trajectories with $u_0, v_0 \sim \mathcal{N}(0,1)$ and $\omega \sim \mathcal{U}(0.5, 50)$ and compute the solution on $[0,1]$ with a spatial resolution of 64.

## C.2 POISSON

We generate a dataset following the $1d$ static Poisson equation eq. (6) with forcing term. For $x \in [0, 1[$,

$$\begin{cases} -\frac{\partial^2 u(x)}{\partial x^2} & = f(x) \\ u(0) & = u_0 \\ \frac{\partial u(0)}{\partial x} & = v_0 \end{cases} \tag{6}$$

We chose $f$ to be a non-linear forcing terms: $f(x) = \frac{\pi}{K} \sum_{i=1}^{K} a_i i^{2r} \sin(\pi x)$, with $a_i \sim \mathcal{U}(-100, 100)$, we used $K = 16$, $r = -0.5$, and solve the equation using a backward finite difference scheme. We generate $1,000$ trajectories with $u_0, v_0 \sim \mathcal{N}(0,1)$ and compute the solution on $[0, 1]$ with a spatial resolution of 64.

## C.3 ADVECTION

The dataset is taken from (Takamoto et al., 2023).

$$\frac{\partial u(t,x)}{\partial t} + \beta \frac{\partial u(t,x)}{\partial x} = 0, \quad x \in (0,1), t \in (0,2] \tag{7}$$

$$u(0,x) = u_0(x), \quad x \in (0,1) \tag{8}$$

Where $\beta$ is a constant advection speed, and the initial condition is $u_0(x) = \sum_{k_i=k_1...k_N} A_i \sin(k_i x + \phi_i)$, with $k_i = \frac{2\pi n_i}{L_x}$ and $n_i$ are randomly selected in $[1, 8]$. The author used $N = 2$ for this PDE. Moreover, $A_i$ and $\phi_i$ are randomly selected in $[0, 1]$ and $(0, 2\pi)$ respectively. Finally, $L_x$ is the size of the domain (Takamoto et al., 2023).

The PDEBench's Advection dataset is composed of several configurations of the parameter $\beta$ ($\{0.1, 0.2, 0.4, 0.7, 1, 2, 4, 7\}$), each of them is composed of $10,000$ trajectories with varying initial conditions. From these datasets, we sampled a total of $1,000$ trajectories for $\beta \in \{0.2, 0.4, 0.7, 1, 2, 4\}$ (which gives about $130$ trajectories for each $\beta$). This gives a dataset with different initial conditions and parameters. Moreover, we sub sampled the trajectories by $4$, leading to a grid of resolution 25 for the t-coordinate and 256 for the x-coordinate.

## C.4 DARCY FLOW

The $2d$ Darcy Flow dataset is taken from (Li et al., 2020) and commonly used in the operator learning literature (Li et al., 2023; Goswami et al., 2022).

$$-\nabla.(a(x)\nabla u(x)) = f(x) \quad x \in (0,1)^2 \tag{9}$$

$$u(x) = 0 \quad x \in \partial(0,1)^2 \tag{10}$$

For this dataset, the forcing term $f$ is kept constant $f = 1$, and $a(x)$ is a piece-wise constant diffusion coefficient taken from (Li et al., 2020). We kept $1,000$ trajectories (on the $5,000$ available) with a spatial resolution is $64 \times 64$.

## C.5 SUMMARY OF PROBLEM SETTINGS CONSIDERED

A summary of the datasets and parameters changing between 2 trajectories are presented in table 5.

## D SOLUTION VISUALIZATIONS

We present here some visualizations from baselines on the different datasets.

| Dataset | Changing PDE data | Range / Generation |
|---------|-------------------|--------------------|
| Helmholtz | $\omega$ 
 $u_0$ 
 $v_0$ | $[0.5, 50]$ 
 $\mathcal{N}(0,1)$ 
 $\mathcal{N}(0,1)$ |
| Poisson | $A_i$ 
 $u_0$ 
 $v_0$ | $[-100, 100]$ 
 $\mathcal{N}(0,1)$ 
 $\mathcal{N}(0,1)$ |
| Darcy | $a(x)$ | $\psi_{\#}\mathcal{N}(0, (-\Delta + 9I)^{-2})$ 
 with $\psi = 12 * \mathbb{1}_{\mathbb{R}_+} + 3 * \mathbb{1}_{\mathbb{R}_+}$ |
| Advection | $\beta$ 
 $A_i$ 
 $\phi_i$ 
 $k_i$ | $\{0.2, 0.4, 0.7, 1, 2, 4\}$ 
 $[0, 1]$ 
 $[0, 2\pi]$ 
 $\{2k\pi\}_{k=1}^8$ |

Table 5: Parameters changed between each trajectory in the considered datasets.

## D.1  HELMHOLTZ

On fig. 3, we present a comparison of baselines on 2 samples. Figure 4 shows the solution through our iterative solver.

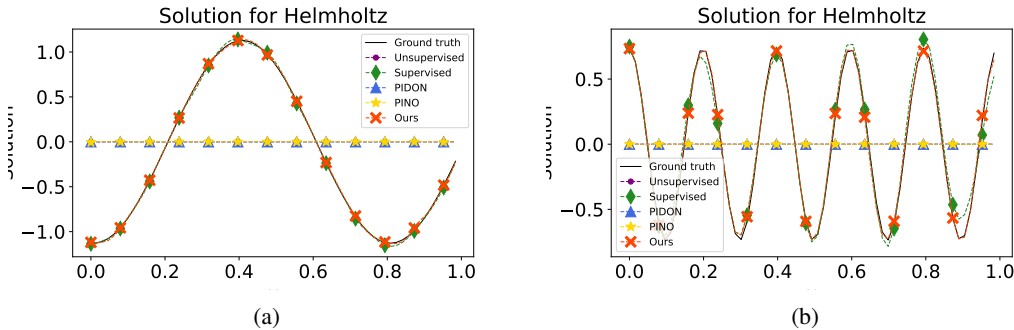

Figure 3: Solutions provided by our solver and baselines on the Helmholtz dataset.

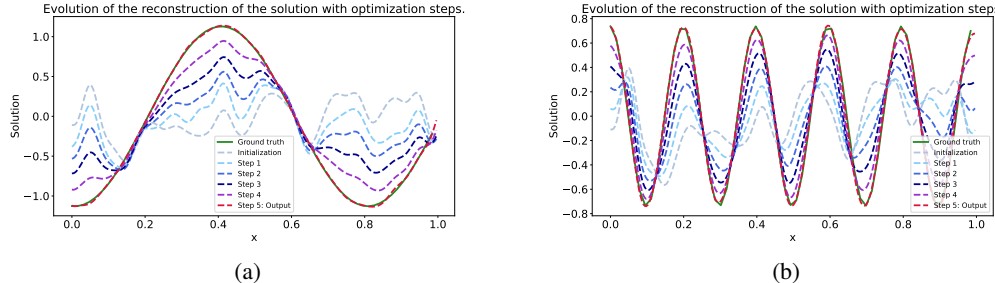

Figure 4: Solution provided by our solver during optimization steps.

## D.2  POISSON

On fig. 6, we present a comparison of baselines on 2 samples. Figure 7 shows the solution through our iterative solver.

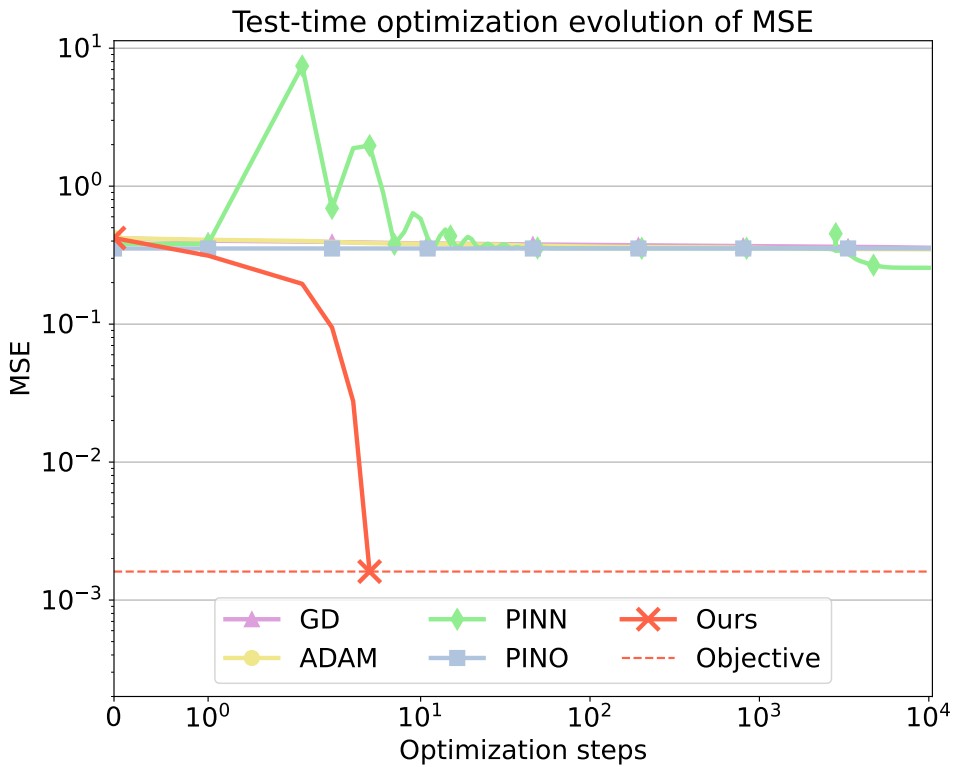

Figure 5: Error vs number of iterations (log-scale). Test-time comparison to other iteration-based approaches on Helmholtz.

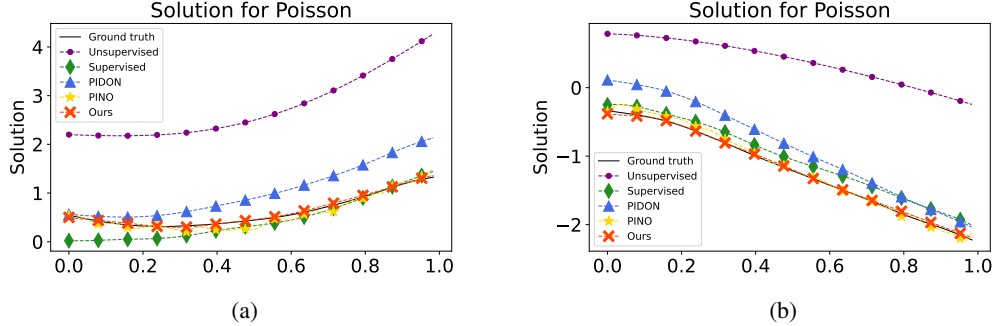

Figure 6: Solutions provided by our solver and baselines on the Poisson dataset.

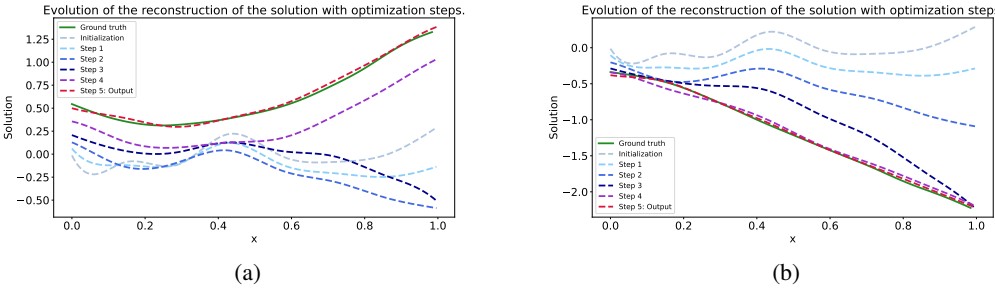

(a)                                                    (b)

Figure 7: Solution provided by our solver during optimization steps.

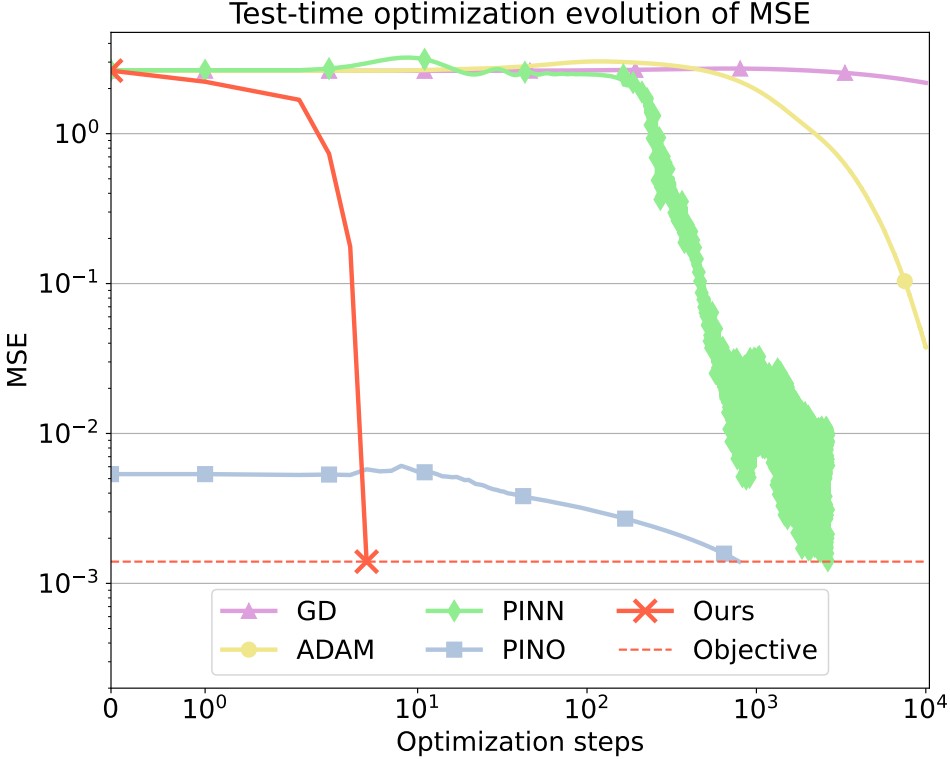

Figure 8: Error vs number of iterations (log-scale). Test-time comparison to other iteration-based approaches on Poisson.

### D.3 DARCY

On fig. 9, we present a comparison of baselines on 2 samples. Figure 10 shows the solution through our iterative solver.

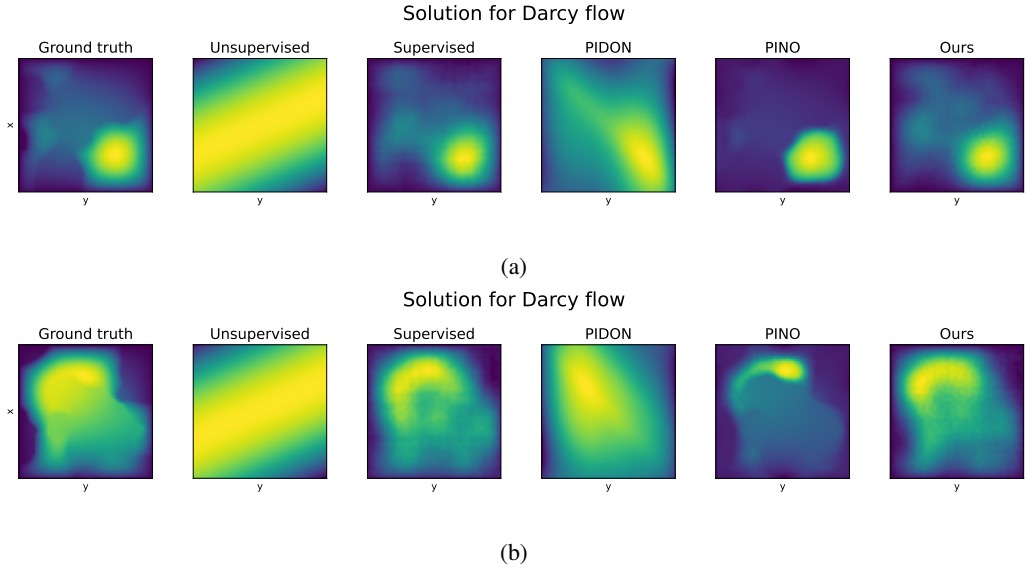

Figure 9: Solutions provided by our solver and baselines on the Darcy dataset.

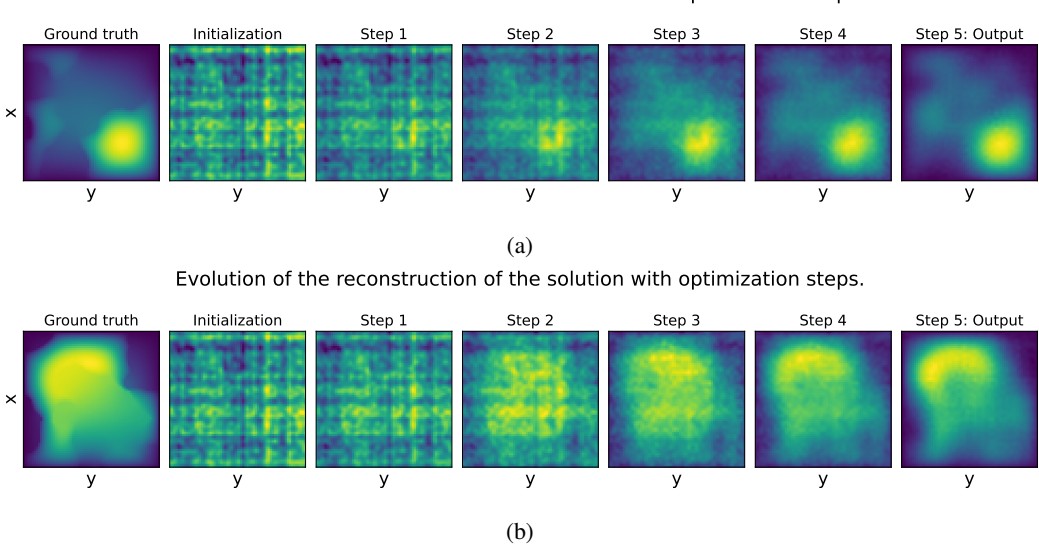

Figure 10: Solutions provided by our solver during optimization steps.

### D.4 ADVECTION

On fig. 12, we present a comparison of baselines on 2 samples. Figure 13 shows the solution through our iterative solver.

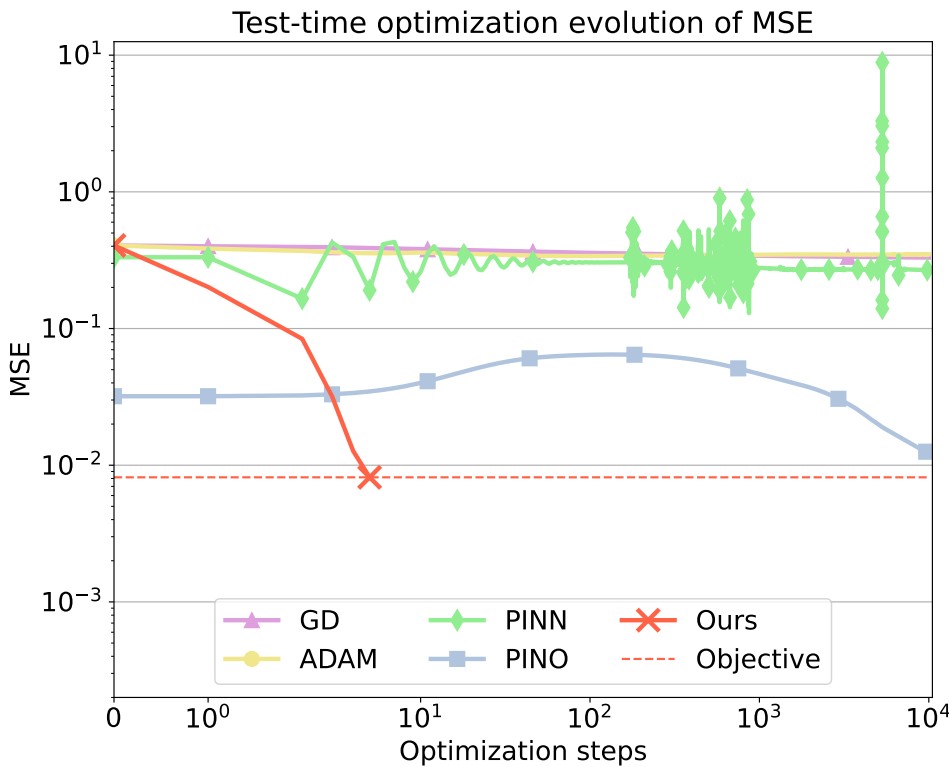

Figure 11: Error vs number of iterations (log-scale). Test-time comparison to other iteration-based approaches on Advection.

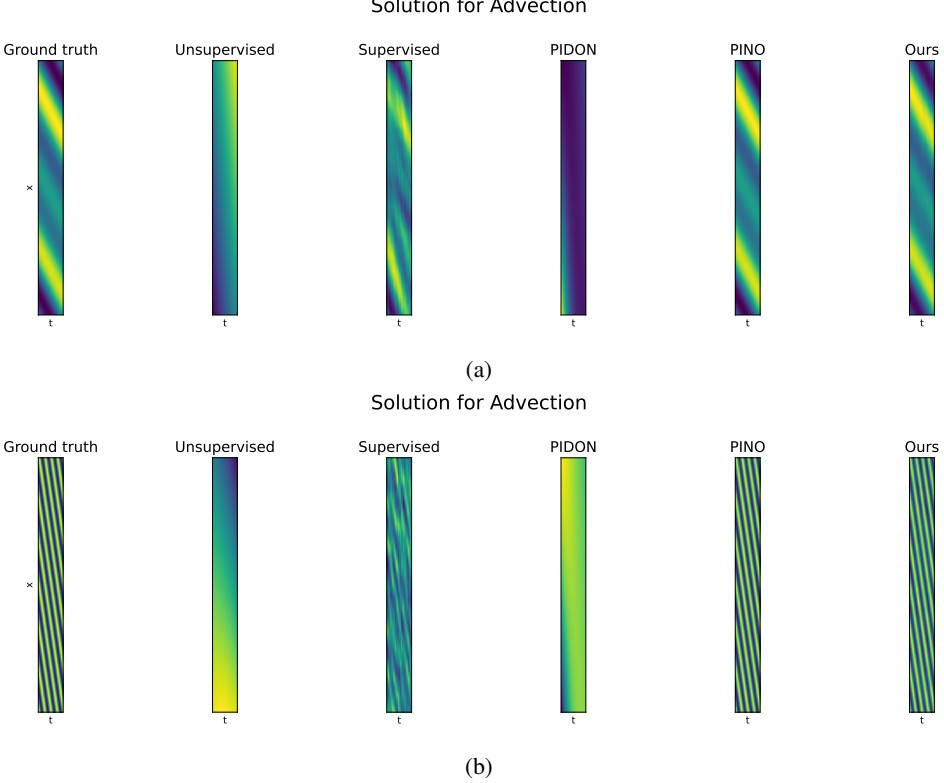

Figure 12: Solutions provided by our solver and baselines on the Advection dataset.

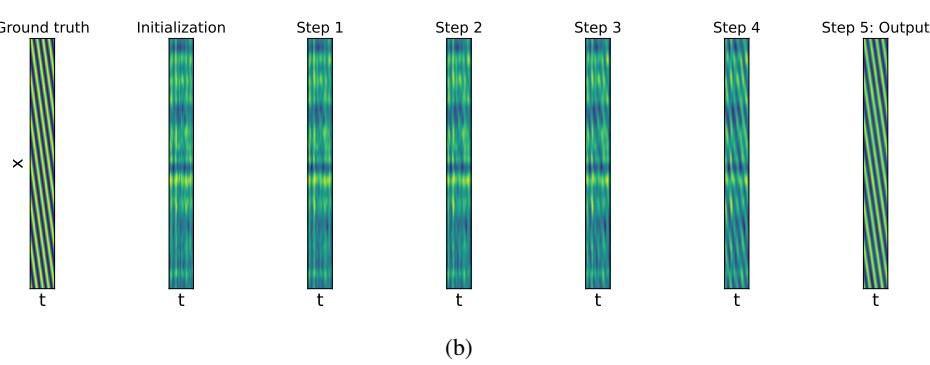

Figure 13: Solutions provided by our solver during optimization steps.

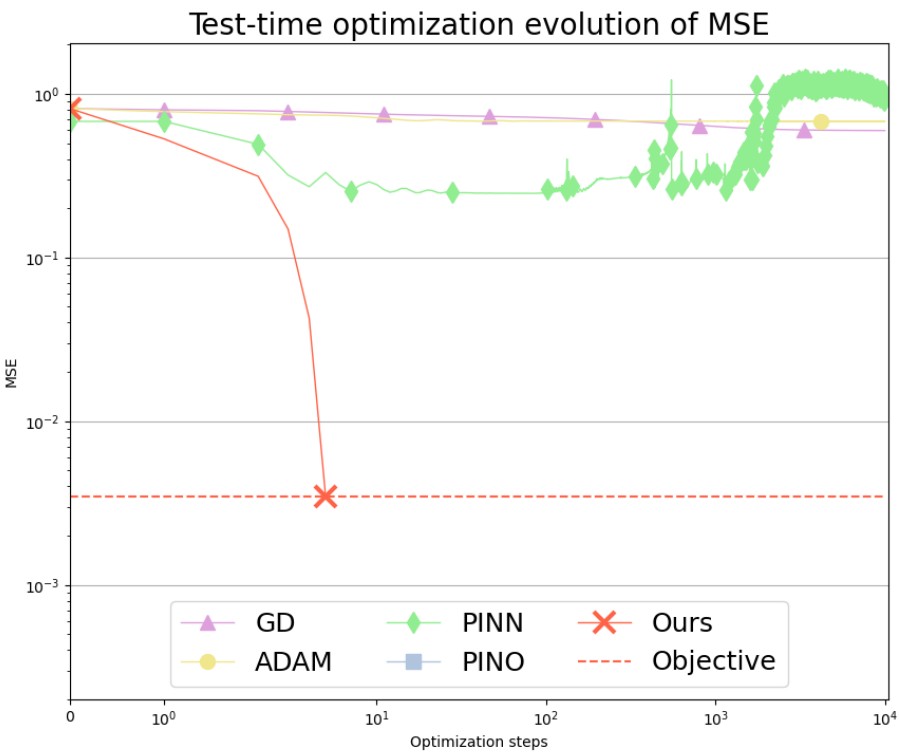

Figure 14: Error vs number of iterations (log-scale). Test-time comparison to other iteration-based approaches on Advection.

