# OpenReview forum: "Learning iterative algorithms to solve PDEs."
_ICLR.cc/2024/Workshop/AI4DiffEqtnsInSci — AI4DiffEqtnsInSci @ ICLR 2024 Poster_

### Official Review · Reviewer_Wx8i · 2024-02-27
**Three major concerns: false claims of novelty, minimal connection to traditional solvers, failure to identify source of empirical gains**

**Rating:** 3
**Confidence:** 5

**Review:**

**Explanation of paper:** This paper solves four benchmark PDEs (1D Poisson, 1D advection, 1D Helmholtz, 2D Darcy flow) using a variation of the physics-informed neural network (PINN) approach. There are two variations compared to the classic PINN approach. (1) Instead of using a neural network as the solution representation, this paper uses a linear sum of B-splines. Thus, the linear coefficients of the B-spline are the parameters to be updated instead of the neural network weights and biases. Splines have previously been used to replace the neural network in a PINN, e.g., [1,2], though it appears the approach here is different. (2) Instead of updating the parameters using a standard optimization algorithm such as gradient descent, the parameters are updated using a *learned optimizer*. This is similar to [3]. Compare page 3 of [3] to equations (3) and (4). Thus, the paper is applying meta-learning to the PINN framework, similar to [4,5,etc]. However, the type of meta-learning is different: previous works appear to have meta-learned loss functions and initial weights, while this work meta-learned the optimizer (see equation (4)).

**Novelty:** Papers do not have to be novel in order to be worthy of acceptance at this workshop. However, this paper explicitly claims that the proposed approach of learning an iterative algorithm to solve a PDE is “novel” and “new”. This is incorrect. Many papers, since at least 2019, have learned an iterative algorithm to solve a PDE. See [6,7,8,9].

Based on a literature review, I believe the novelty of this paper is in meta-learning an optimizer for the PINN framework of optimizing a PDE residual. It is possible that spline basis functions have not been used to replace neural network basis functions in the PINN framework, but I am not sure.

**Connection to traditional numerical solvers:** The first two paragraphs of section 2.2 explain the connection between traditional numerical solvers and this method. However, there are multiple incorrect statements in these paragraphs. The first sentence of section 2.2 says that
> Traditionally, numerical solvers ... iteratively [update] parameters $\theta$ based on the minimization of some criterion $\mathcal{L}_{PDE}$ (e.g. the PDE residual)

Iterative numerical solvers do not use the PDE residual to update the parameters $\theta$. Instead, they use a *weak* form of the PDE, derived by multiplying the PDE by each of the basis functions and integrating. This is an important difference, because it leads to an entirely different solution methodology. Using the weak form of the PDE leads to linear (or sometimes non-linear) systems of equations, while using the PDE residual leads to a non-linear optimization problem. Another incorrect statement is that traditional numerical solvers are iterative. Some traditional numerical solvers are iterative, but many (perhaps most) are not. Iterative numerical methods are used for elliptic PDEs, but hyperbolic and parabolic PDEs do not use iterative updates. A third incorrect statement is in the third sentence of section 2.2:
>As opposed to traditional methods, the iterative solver $\mathcal{A}$ is not specifically tailored or handcrafted to the given problem, but learned from the data.

Traditional iterative numerical solvers are not always tailored or handcrafted to the given problem. In some cases, they are. The Multigrid method is one example. In many cases, however, the goal is to solve a linear system of equations. Any method of solving a linear system (such as LU decomposition, for example) or non-linear system (such as Newton’s method) can be used.

While these incorrect statements are not essential to the correctness of the paper, they reveal that the connection with traditional numerical solvers is not nearly as strong as the paper claims.

Essentially the only connection between this method and traditional iterative numerical solvers is that they are both iterative.

**Failure to identify the source of empirical gains:** Please read two papers: *Troubling trends in machine learning scholarship”* [10] and *Winner's curse: on pace, progress, and empirical rigor* [11]. [10] discusses four troubling trends, the second of which is
> Failure to identify the sources of empirical gains, e.g. emphasizing unnecessary modifications to neural architectures when gains actually stem from hyper-parameter tuning.

[11] discusses the same issue using different terminology:
> Looking over papers from the last year, there seems to be a clear trend of multiple groups finding that prior work in fast moving fields may have missed improvements or key insights due to things as simple as hyperparameter tuning studies or ablation studies.

[11] recommends standards for empirical evaluation that "should be encouraged, rewarded, and ultimately required in empirical work". The third recommendation is
> **Ablation Studies** Full ablation studies of all changes from prior baselines should be included, testing each component change in isolation and a select number in combination.

This paper claims that the empirical gains over baselines (see figure 1a) are from learning an iterative update (see equation (4)). This may be true. However, this paper does not provide enough evidence to deduce that it is true; it is possible that other factors (such as using B-splines) may be the source of empirical gains. To identify the source of empirical gains, an ablation study should be performed which tests each "component change" (meta-learning via equation (4) and using B-spline basis functions instead of neural networks) in isolation. It is possible that a B-spline basis representation alone would be sufficient to achieve the performance seen in figure 1a; it is also possible that using a neural network basis representation while learning an iterative update would also be sufficient to achieve the performance seen in figure 1a. Without an ablation over these component changes, it is impossible for the reader to determine what the source of empirical gains is.

**Additional comments:**
* Given that (a) learning to solve PDEs is not a novel idea, and (b) the iterative algorithms being learned are entirely different from iterative algorithms in traditional numerical solvers, the title is inappropriate and misleading. Changing the title should be a necessary (but not sufficient) condition for acceptance of this paper. I suggest changing the title to “Meta-learning the optimizer of physics-informed B-spline basis functions to solve PDEs”.
* In certain respects (such as the title, abstract, overall framing) I find the paper to be ostentatious (i.e., designed to attract attention rather than to accurately report the ideas and findings).
* Iterative methods are not typically used on the advection equation. I’m not sure using this as a benchmark problem makes much sense.
* An advection problem with two dimensions (space, time) is usually considered 1D, not 2D.
* Is the Helmholtz equation being solved in figure 1b? I think it is, because figure 1b is identical to figure 4b in the appendix. In any case, you should clarify that in the main text.
* What PDE is being solved in figure 1a?
* Why aren’t plots like figure 1a included for every benchmark PDE?
* ODIL [12] is a much stronger baseline than PINN (and possibly PINO) for these PDEs.


**Conclusion:** I think the concept presented in this paper -- meta-learning an optimizer for PINNs and/or physics-informed spline basis functions -- is of interest to the workshop, and could be worthy of acceptance. However, I have serious concerns about the accuracy of the paper. The paper claims to present a novel approach, but this is false. The paper claims to be inspired by traditional numerical methods, but essentially the only connection between the two methods is that one is iterative and the other can sometimes be iterative. The paper also combines two component changes without performing any ablation study, meaning that they failed to identify the source of empirical gains. I think this paper should be rejected.

Should future versions of this paper correct the three main concerns listed in the paragraph above, I would conclude that this paper would be deserving of acceptance in this workshop.

[1] Wandel, Nils, et al. "Spline-pinn: Approaching pdes without data using fast, physics-informed hermite-spline cnns." Proceedings of the AAAI Conference on Artificial Intelligence. Vol. 36. No. 8. 2022.

[2] Doległo, Kamil, et al. "Deep neural networks for smooth approximation of physics with higher order and continuity B-spline base functions." arXiv preprint arXiv:2201.00904 (2022).

[3] Li, Ke, and Jitendra Malik. "Learning to optimize." arXiv preprint arXiv:1606.01885 (2016).

[4] Penwarden, Michael, et al. "Physics-informed neural networks (PINNs) for parameterized PDEs: a metalearning approach." Available at SSRN 3965238 (2021).

[5] Qin, Tian, et al. "Meta-PDE: Learning to Solve PDEs Quickly Without a Mesh." arXiv preprint arXiv:2211.01604 (2022).

[6] Greenfeld, Daniel, et al. "Learning to optimize multigrid PDE solvers." International Conference on Machine Learning. PMLR, 2019.

[7] Luz, Ilay, et al. "Learning algebraic multigrid using graph neural networks." International Conference on Machine Learning. PMLR, 2020.

[8] Zhang, Enrui, et al. "A hybrid iterative numerical transferable solver (HINTS) for PDEs based on deep operator network and relaxation methods." arXiv preprint arXiv:2208.13273 (2022).

[9] Hsieh, Jun-Ting, et al. "Learning neural PDE solvers with convergence guarantees." arXiv preprint arXiv:1906.01200 (2019).

[10] Lipton, Zachary C., and Jacob Steinhardt. "Troubling trends in machine learning scholarship." arXiv preprint arXiv:1807.03341 (2018).

[11] Sculley, David, et al. "Winner's curse? On pace, progress, and empirical rigor." (2018).

[12] Karnakov, Petr, Sergey Litvinov, and Petros Koumoutsakos. "Solving inverse problems in physics by optimizing a discrete loss: Fast and accurate learning without neural networks." PNAS nexus 3.1 (2024): pgae005.

---

### Official Review · Reviewer_ysCE · 2024-02-29
**incremental work, but interesting. need to add relevant work on unrolled neural networks for solving inverse problems**

**Rating:** 6
**Confidence:** 4

**Review:**

This submission deals with learning PDEs using a recurrent neural network inspired by iterative solvers. Experiments with classical PDEs show the advantage of this method compared with existing methods such as PINN and PINO.

This is an important problem and very useful to speed up high-dimensional PDEs for simulations. The idea is natural and incremental. Also, the paper misses an important connection with existing work on “unrolled neural networks” for solving inverse problems that appear for example in imaging, see for example [1]. Unrolled neural networks also rely on unrolling iterations of classical optimization methods such as proximal gradient descent.

[1] Mardani, M., Sun, Q., Donoho, D., Papyan, V., Monajemi, H., Vasanawala, S., & Pauly, J. (2018). Neural proximal gradient descent for compressive imaging. Advances in Neural Information Processing Systems, 31.

more comments
Do you share the weights of the neural network across all iterations? What if you let the weights be independent across iterations. The results from unrolled neural networks were observing improved performance by variable weight training.

---

### Official Review · Reviewer_c2E1 · 2024-03-01
**Promising results in the field on hybrid models**

**Rating:** 7
**Confidence:** 3

**Review:**

**Summary**: The manuscript discusses a novel hybrid method to solve PDEs, which lies at the intersection of PINNs and an operator-based approach. The authors combine iterative methods for PDEs to learn the optimization itself, coupled with supervised learning strategies. Optimistic results are showcased on conventional PDEs compared with existing approaches.


**Strengths**
1. The paper is easy to read and follow. The authors describe their approach succinctly, and the hybrid algorithm is a novel contribution.
2. The problem seems well-motivated concerning the pitfalls of using PINNs or operator-based approaches.
3. The authors showcase initial promising results and good details about the experiments and setup.

**Areas of improvement**:

1. I have some concerns regarding the performance evaluations. No. of iterations do not provide a clear idea of how great a method is performing computationally. In hindsight, the devised approach would have more inference time than a single forward pass used in PINOs and PINNs. Hence, reporting the computational time of training and testing periods will make the argument more convincing.
3. The problems chosen for benchmarks are quite simple: very fast FEM-based solvers exist, which have been shown to beat these approaches by orders of magnitude [grossman2023, markdis2021]. However, PINNs and PINOs are useful in high dimensional PDEs and non-local problems where we do not have good classical numerical solvers. Evaluating this set of problems will make the argument for these methods more convincing.

[grossmann2023] Grossmann, Tamara G., et al. "Can physics-informed neural networks beat the finite element method?." arXiv preprint arXiv:2302.04107 (2023).

[markidis2021] Markidis, Stefano. "The old and the new: Can physics-informed deep-learning replace traditional linear solvers?." Frontiers in Big Data 4 (2021): 669097.

---

### Meta-Review · Area_Chair_HUh6 · 2024-02-28

**Recommendation:** Accept (Poster)

**Metareview:**

This paper proposes an iterative approach to learning PDEs. I agree with Reviewer Wx8i that the authors should clarify their comments on novelty and correct some of the statements on the connection to numerical methods. However, the iterative form is similar to the semi-discrete form in numerical methods, which I think is what the authors meant and is a nice connection.  The authors should also add the suggested references and the unrolled neural network reference from Reviewer ysCE. With these revisions, I vote for acceptance to the workshop.

---

### Decision · Program_Chairs · 2024-02-28

Accept (Poster)